# Assessment of the Quality, Chemometric and Pollen Diversity of *Apis mellifera* Honey from Different Seasonal Harvests

**DOI:** 10.3390/foods12193656

**Published:** 2023-10-03

**Authors:** Andrés Rivera-Mondragón, Maravi Marrone, Gaspar Bruner-Montero, Katerin Gaitán, Leticia de Núñez, Rolando Otero-Palacio, Yostin Añino, William T. Wcislo, Sergio Martínez-Luis, Hermógenes Fernández-Marín

**Affiliations:** 1Centro de Investigaciones Farmacognósticas de la Flora Panameña (CIFLORPAN), Departamento de Química Medicinal y Farmacognosia, Facultad de Farmacia, Universidad de Panamá, Panama City P.O. Box 3366, Panama; andres.riveraa@up.ac.pa; 2Instituto Especializado de Análisis (IEA), Universidad de Panamá, Panama City P.O. Box 3366, Panama; katerin.gaitan@up.ac.pa (K.G.); leticia.denunnez@up.ac.pa (L.d.N.); 3Sistema Nacional de Investigación (SNI), Senacyt, Ciudad del Saber, Clayton 0843-03081, Panama; gbrunerm@gmail.com; 4Centro de Biodiversidad y Descubrimiento de Drogas, Instituto de Investigaciones Científicas y Servicios de Alta Tecnología (INDICASAT-AIP), Ciudad del Saber, Clayton 0843-01103, Panama; mmarrone@indicasat.org.pa; 5Coiba Scientific Station (Coiba AIP), Gustavo Lara Street, City of Knowledge, Clayton 0843-01853, Panama; yostin0660@gmail.com; 6Facultad de Agronomía, Universidad de Panamá, Panama City P.O. Box 3366, Panama; rolando.otero@up.ac.pa; 7Museo de Invertebrados G. B. Fairchild, Universidad de Panamá, Panama City P.O. Box 00017, Panama; 8Smithsonian Tropical Research Institute, Apartado 0843-03092, Panama; wcislow@si.edu

**Keywords:** *Apis mellifera*, Panama, physicochemical analysis, seasonal variation, chemometrics

## Abstract

The parameters for assessing the quality of honey produced by *Apis mellifera* are standardized worldwide. The physicochemical properties of honey might vary extensively due to factors such as the geographical area where it was produced and the season in which it was harvested. Little information is available on variations in honey quality among different harvest periods in tropical areas, and particularly in neotropical dry forests. This study describes variations in seventeen physicochemical parameters and the pollen diversity of honey harvested from beehives during the dry season in February, March, and April 2021, in the dry arc of Panama. Potassium is the most abundant mineral in honey samples, and its concentration increases during the harvest period from February to April. A PCA analysis showed significant differences among the samples collected during different harvest periods. The pollen diversity also differs among honey samples from February compared with March and April. The results indicate that climatic conditions may play an important role in the quality of honey produced in the dry arc of Panama. Furthermore, these results might be useful for establishing quality-control parameters of bee honey produced in Panama in support of beekeeping activities in seasonal wet–dry areas of the tropics.

## 1. Introduction

Honey is a viscous and sweet liquid that is naturally produced by social honey bees of the genus *Apis* and by the stingless bees of the tribe Meliponini [1]. Bees collect nectar from flowers or extrafloral nectaries and transport it to their beehives, where it is stored in the combs of the hive and transformed by reducing the water content and adding other materials [2]. Honey is used by cultures around the world without any processing (e.g., purification or heating) to make it suitable for human consumption [3].

Currently, the principal honey used for human consumption is produced by bees of the genus *Apis*, particularly *Apis mellifera*. Honey is a mixture of sugars, typically containing around 40% fructose, 30% glucose, and approximately 5% sucrose [3]. In addition, honey contains small amounts of phenolic compounds (60 to 460 mg/100 g) [4], minerals (<0.6%) [5], free amino acids (1%) [6], and proteins (0.1–0.5%) [7]. Honey also contains vitamins and essential oils; however, the presence of these components is variable and is related to the botanical origin of honey [8]. The composition of honey is determined by various factors including the variety of the honey bees, the species of flowers used, and the geographical region where it is produced [9]. The physical–chemical characteristics of honey also are influenced by the honey’s processing and storage time [10] and also the pre- and postharvest management by beekeepers [11].

Honey intended for human consumption should be of high quality, safe, and free of contamination and defects. Analytical methods for the quality control of honey—from the hive to the final consumer product—are described in publications by the International Honey Commission (ICH) and other organizations [2,5,8]. These methods include chromatography [3], spectrophotometry [7], and refractometry [8]. As new and better technologies become available, introducing and updating new standards using new analytical and statistical methods will be necessary. Honey is one of the most frequently altered natural food products; honey can be mislabeled with regard to its botanical and geographical origins, be adulterated by mixing floral honey with other resources that bees might collect, such as sugar water, or by adding flavored sugar solutions [12]. The properties of honey can also change due heating or storage in unsatisfactory conditions. For example, hydroxyl methyl furfural (HMF) is a product of the degradation of sugars, such as fructose and glucose, which occurs during honey’s processing and long-term storage [10]. The concentration of HMF indicates the freshness of honey, because it is typically absent (or present only in very small amounts) in fresh honey and tends to increase during processing and/or because of aging. Therefore, a high concentration of HMF is indicative of poor storage conditions and/or the excessive heating of honey [10]. Moreover, pollen found in honey is used to determine possible floral resources that honey bees visit, though the collection of pollen and nectar is often conducted by different bees within a colony [13,14].

The aim of this study was to characterize the physical and chemical properties of honey, including the concentrations of macrominerals (K, Na, and Ca), trace metals (Pb, Cd), and the pollen diversity of three different harvests, using Africanized bees, *Apis mellifera*, from beehives located in a bee apiary in a relatively dry region of tropical Panama.

## 2. Materials and Methods

### 2.1. Honey Samples

The study was carried out using bees in the “arco seco” (dry arc) region of central Panama near the Pacific Ocean, a narrow belt of extremely low precipitation (1400 mm of rain per year) [15]. Beehives of the INDICAST-AIP apiary were used, located at the National Institute of Agronomy (Divisa District, Herrera Province; 8°08′13.3″ N 80°42′05.5″ W), surrounded by a riparian matrix of the Santa María River; experimental plots, including pasture for cattle, rubber, and mango plantations; small plantations for educational use; and large extensions of reedbeds. The apiary has 21 hives, but only 16 hives produced honey in at least one harvest (Appendix A).

Samples of honey produced by beehives were collected three times during the honey-production season, which occurs during the dry season (~late December to late April [16,17]: 20 February, 24 March, and 30 April 2021. For each harvest, all hives were harvested on the same day. The frames were placed in boxes for transportation and labeled with the hive number. The frames selected for harvesting had >75% of their honey cells sealed. The honey samples were centrifuged with a piece of four-panel equipment, and the honey from individual beehives was filtered through a 5 mm sieve, then a 0.5 mm sieve, and stored in a five-gallon container. Smaller subsamples of honey were poured into 750 mL plastic containers, sealed with a screw cap, and stored at room temperature (for 10 months) until analyzed. All honey samples were used in their raw form, and they were neither pasteurized nor subjected to other thermal treatments.

### 2.2. Physicochemical Analysis

The following physicochemical properties were determined according to standard protocols, as recommended by the International Honey Commission (IHC) and related standards [2,5,7,8]: pH, conductivity, salinity, total dissolved solids (TDS), °Brix, moisture, acidity, HMF concentration, ash content, and mineral and pollen compositions.

#### 2.2.1. pH, Electrical Conductivity, Total Dissolved Solids (TDS), and Salinity

The pH of the samples was determined according to the IHC [18]. Briefly, 10 g of honey was diluted with 75 mL of ultrapure water (Millipore Water, Direct-Q^®^ 5 UV) in a 100 mL beaker. The mixture was shaken for 10 min using a magnetic stirrer at 250 rpm. A multiparameter analyzer (ST3100M-N, Ohaus, Parsippany, NJ, USA) equipped with a pH electrode (ST230, Ohaus) was used to measure the pH, and a conductivity electrode (STCON3, Ohaus) was used for the analysis of the electrical conductivity, total dissolved solids, and salinity, expressed in units of µS/cm, mg/L, and psu, respectively.

#### 2.2.2. Moisture and Total Soluble Solids (°Brix)

The moisture and total soluble solids parameters (°Brix) were determined with a refractometer (ATC, Aichose, China). The samples were directly analyzed to measure the refractive index and °Brix at 20 °C.

#### 2.2.3. Determination of Free Acidity, Lactones, and Total Acidity

Briefly, 10 g of honey was dissolved in 75 mL of ultrapure water, and the determination of the free acidity, lactones, and total acidity followed the methodology of Bogdanov (2009) [18].

#### 2.2.4. Hydroxy Methyl Furfural (HMF) Determination

Following the IHC’s guidelines, the HMF was measured using White’s spectrophotometric method [8,18,19]. The absorbance measurements at 284 and 336 nm were measured in a UV–Vis spectrometer (Shimadzu, UV-2401PC). The HMF content was calculated (in mg/kg), as follows [18]:HMF (mg/kg) = (Abs284–Abs336) × 149.7 × 5 × dilution factor/g of honey

#### 2.2.5. Determination of Ash and Mineral Composition

Approximately 10 g of honey was placed in a previously weighed porcelain crucible and heated on a hotplate until completely dry (from 100 to 250 °C for at least 8 hours). The crucibles with the samples were then introduced in muffle ovens and calcinated to ash at 500 °C (method 1) or 450 °C (method 2). All samples were analyzed in triplicate. The resulting ashes were weighed. The ash content of honey is expressed as a percentage by weight (*w/w* %).

White ashes obtained with method 1 were then dissolved in a mixture of equal parts of HCl 2 M and HNO_3_ 2 M and diluted with ultrapure water in a 100 mL volumetric flask [20]. The resulting solution was then used for Na determination. For the analysis of K, 5 mL of the previous solution was transferred to a volumetric flask and diluted with deionized water to 50 mL. For the determination of the Ca content, a lanthanum oxide solution (Merck, Germany) (100 g/L) was added to the samples so that the final solution contained 1% of this reagent. White ashes obtained with method 2 were then dissolved in 5% HNO_3_ (trace metal grade, Sigma) and diluted to a volume of 50 mL with the same solution [21]. The solution was used to determine Pb and Cd.

The concentration of minerals in the honey samples was determined using atomic absorption spectrometry. Sodium (Na), potassium (K), calcium (Ca), lead (Pb), and cadmium (Cd) were determined with a Shimadzu AA-7000 Atomic Absorption Spectrophotometer equipped with an air–acetylene flame atomizer. The mineral content was read from a calibration curve plotted from calculated points (Table 1). The analytical accuracy and precision were within 5% for all elements. The content of minerals in the honey samples is expressed in µg/g (wet mass). All samples were analyzed in triplicate.

### 2.3. Acetolysis and Pollen Identification

To determine which flowers the bees visited for pollen collection, we analyzed nine samples from three harvests. From each sample, we used 50 mL of honey bee (beehive/harvest) for acetolysis. To remove the sugar, the honey was gradually decanted into a 500 mL chemical glass with 450 mL of water and shaken. The samples were placed in 50 mL plastic falcon tubes with 50 mL of solution and centrifuged at 5000 rpm. The final amount of water varied, since the honey differed in density; the water was carefully decanted, and the sediment (i.e., pollen) was retained. This step was repeated three times. The samples were stored at room temperature. The acetolysis followed general methods for palynology [22], and the pollen was identified to the morphospecies using those provided in Roubik and Moreno [23]. The pollen grains were photographed, measured, and identified to the level of species, genera, family, or undetermined.

### 2.4. Statistical Analyses

All data analyses were performed using the R v.4.2.0 statistical package [24]. Descriptive statistics of the physicochemical parameters were calculated for each hive and harvest period. Pb and Cd were excluded from all analyses, as they were not detected in the samples. The normality of the data was tested before the analysis using the Shapiro–Wilk test with the ‘shapiro_test()’ function. To determine the association among the physicochemical parameters, a correlation matrix was calculated and visualized using the ‘corrplot()’ function in the ‘corrplot’ package [25]. The data were arranged in a matrix including 27 rows (replicates) and 17 columns (15 physicochemical parameters as numerical variables, and beehive-ID and harvest period as categorical variables).

Factor analysis of mixed data (FAMD), a principal component method that incorporates quantitative and qualitative variables, was employed to explore the similarity among the physicochemical parameters across the harvest periods [26]. In the FAMD analysis, missing values (NAs) were imputed using the ‘missMDA’ package [27]. Because some of the beehives did not produce enough honey for the physicochemical analysis, incomplete data were tabulated as NA. To handle missing data, we used the ‘missMDA’ package, which imputes missing values in a way that assigns no weight to the imputed values during the multivariable analysis using the iterative PCA algorithm. Subsequently, hierarchical clustering on principal components (HCPC) was conducted to cluster similar physicochemical parameters based on the reduced number of variables determined with the FAMD analysis [26]. The number of factors or dimensions to retain in the FAMD analysis was determined using the Kaiser and Guttman criterion [28], which retains eigenvalues greater than 1. For the analysis, the physicochemical variables were normalized at the same scale using the scale function. The ‘FAMD()’ and ‘HCPC()’ functions from the ‘FactoMineR’ package [29] were utilized for the FAMD and HCPC analyses, respectively [30]. To visualize the proportion of the variation retained by each principal component, we examined the eigenvalues using the ‘fviz_eig()’ function in the ‘factoextra’ package [30]. We used the ‘dimdesc()’ function in the ‘FactoMineR’ package [29] to determine the significant contribution of each physicochemical parameter in the analysis. Finally, we used the ‘ggplot’ function in the package ‘ggplot2′ to visualize the scores [31].

To examine the differences among physicochemical parameters across harvest periods, a Kruskal–Wallis test was performed using the ‘kruskal.test()’ function. Following the Kruskal–Wallis test, a post hoc analysis was conducted using Dunn’s test with Bonferroni correction, which was implemented using the ‘dunnTest()’ function from the ‘FSA’ package. Missing values were removed prior to conducting these analyses. A similar test was employed to investigate the relationship between the physicochemical parameters and beehive-ID. However, the results revealed nonsignificant differences between the physicochemical parameters and beehive-ID (*p* ≥ 0.05, for all parameters). Therefore, the beehive-ID factor was, subsequently, excluded from our analysis.

To evaluate the differences in the pollen species diversity among the harvest periods, a distance matrix was calculated using the ‘vegdist()’ function from the ‘vegan’ package, using the Jaccard dissimilarity index [32]. The calculated distance matrix was then used to perform a hierarchical cluster analysis using the ‘hclust()’ function, applying Ward’s method.

## 3. Results and Discussion

### 3.1. Physicochemical Analysis

#### 3.1.1. Moisture

The water content of honey may have an important impact on its microbiological stability and shelf life, by changing the likelihood of fermentation. Honey that has a high water content may facilitate the fermentation process. As shown in Appendix A, the water content of honey ranged from 16.5% in February to 22.5% in April. International standards recognize that high-quality honey should have a moisture content under 20 g/100 g (20% *w/w*) [2,33]. Panamanian legislation sets the moisture content ≤ 18.5% [34]. All samples from February to March (except for two samples: 21.0 and 23.0%) followed the moisture content limits, while all honey harvested during April exceeded the maximum amount allowed, as the weather was becoming more humid. The last samples were collected at the beginning of the rainy season in Panama, when the nectar likely has a greater water content, and increasing humidity may slow the dehydration of the honey in the hive.

#### 3.1.2. Free Acidity, pH, Lactone Acidity, and Total Acidity

The results of the pH measurements are depicted in Table 2, showing that all honey samples were in the acidic range. In turn, the highest pH value was observed among the honey samples collected in April, at 4.60, and in February for the lowest pH value, reaching 3.92 (see Appendix A). Levels for the pH value are not usually defined as a quality standard, and the pH values of the different kinds of honey usually range from 3.5 to 5.5 [35]. This parameter might be relevant during storage, since a pH value should be low to inhibit the growth of microorganisms, as the optimum pH for most microorganisms is between 7.2 and 7.4 [36,37].

As shown in Appendix A, the free acidity of the evaluated honey varied between 26.2 meq/kg and 72.2 meq/kg. According to the standards [1,32], the permissible value of free acidity in honey is 50 meq/kg. Panamanian legislation set 40 meq/kg as the maximum free acidity content [34]. Nine samples harvested in March and April exceeded the limits of international regulations. Honey acidity is linked with the natural presence of organic acids, mainly gluconic acid and their lactones, esters and some inorganic ions (such as phosphates, sulfates and chlorides) [35,38]. Free acidity could be used as an indicator of honey deterioration. The high acidity in honey samples can be due to the fermentation of sugars into organic acids [35,36].

The determination of the lactone acidity content, considered to be an acidity reserve when the honey becomes alkaline, is of interest, since its hydrolysis induces a rise in free acids [38]. Our results indicate that the lactone acidity level ranged from 12.7 to 43.7 meq/kg. The total acidity, which is the sum of free and lactone acidities, ranged from 35.0 to 102.0 meq/kg. Standards are not set for any lactone or total acids for different kinds of honey.

#### 3.1.3. Ash Content and Electrical Conductivity

Our results showed that the ash content ranged from 0.14 to 0.57% (*w/w*) (Appendix A). The average ash content for April (0.42%) was the highest value observed in comparison to the ash percentages obtained for the other two collecting periods (Table 2). The ash content is used as an indirect measurement of the minerals present in honey, and it ranges between 0.02 and 1.03% (*w/w*) [37]. Although international standards are not given for ash content, quality standards in Panama recommend a maximum value of 0.60% (*w/w*) [34].

As the electrical conductivity of honey is related to the ash content, it was recently incorporated into the Codex Alimentarius standards, replacing the quantification of the ash in honey [37]. As shown in Appendix A, the electrical conductivity values varied between 236 and 939 µS/cm. Similar to the ash content, the highest average value was obtained in April, and the lowest electrical conductivity averages were found among samples collected in February (Table 2). All honey samples were below the maximum limit of 800 µS/cm recommended by international guidelines, except for one sample from March (939 µS/cm) and one sample from April (914 µS/cm).

#### 3.1.4. Total Soluble Solids (°Brix)

The results of the total soluble solids are presented in Appendix A, showing that the °Brix values ranged from 75.5 to 82.5. All samples evaluated were similar to values reported in earlier studies [39,40], which ranged between 72.2 and 82.8 °Brix. Current Panamanian legislation does not establish values for total soluble solids.

The average value obtained in April (77.3 °Brix) was slightly lower than those obtained in February and March (≈80 °Brix) (Table 2). In general, for most of the samples collected in February and March, the total soluble solids were generally more than 80 °Brix and can be considered of high grade and highly stable upon storage. On the other hand, the honey collected in April with less than 80 °Brix soluble solids are more likely to be fermented during storage.

#### 3.1.5. Hydroxy Methyl Furfural (HMF) Content

HMF is recognized as a freshness indicator of honey, which is produced by the decomposition of fructose (dehydration in an acid medium) when honey is submitted to a long storage time or after heat treatment [8,37,38]. As shown in Appendix A, the amounts of HMF in the honey samples varied from 22.0 to 61.5 mg/kg. Eighteen out of thirty-three honey samples were above the standard limit set by Panama (40 mg/kg) [34]. However, honey produced in subtropical regions can contain higher HMF concentrations without overheating or adulteration due to the normally high local temperatures [41]. International standards allow for a maximum HMF of 80 mg/kg in the case of honey of declared origin from countries or regions with tropical ambient temperatures (including Panama). If this last specification was adopted by Panama as a new quality standard, all honey samples would be below the maximum limit and, therefore, satisfy this criterion. The HMF concentration is widely used to monitor the quality of honey, and it is commonly absent or present in very small amounts in fresh honey, while its concentration tends to increase during poor storage conditions or excess heating.

#### 3.1.6. Mineral Content

Data from the results of the minerals determined in the honey samples are summarized in Table 2 and Appendix A. K was the most abundant mineral determined in the honey samples, ranging from 307.2 to 1955.0 µg/g, followed by Ca (58.5–248.1 µg/g) and Na (14.7–75.9 µg/g). The mineral content can be used as a parameter to evaluate the nutritional value of honey. Minerals are described as being important in the human diet, as they play an essential role in the transmission of stimuli in muscles and nerves, in the transmission of intracellular signals, and as cofactors for hundreds of enzymes involved in the metabolic functions of the human body [42,43].

The mineral content can also be indicative of environmental pollution, especially the presence of Cd, Pb, and other heavy metals [44], which may present hazards to human health and adversely affect the quality and safety of honey. In addition, the quantification of these trace toxic elements can serve as bio-indicators for heavy metal contamination. Honey is known to accumulate trace metals, including heavy metals such as Pb and Cd. Thus, the determination of the level of toxic heavy metals in honey samples is an important indicator of environmental pollution. The levels of Pb and Cd in all honey samples were below the limit of detection, which were less than 1.25 µg/g and 0.125 µg/g (wet mass), respectively. Our results suggest the presence of low levels of contamination in the soil and flora in the surrounding areas of the apiary. This result is consistent with the fact that the samples of this study were collected from hives located far from main roads, factories, or urban areas.

Although honey should be free from heavy metals in amounts that may represent a hazard to human health, maximum residue levels of these potentially toxic elements in honey have not been established [8]. According to WHO guidelines, the provisional tolerable weekly intake (PTWI) for Pb and Cd is 210 µg/day for a 60 kg adult person [45]. Based on the results obtained in this investigation, consuming 20 g per day of honey would provide less than 25 and 2.5 µg of Pb and Cd, respectively, per day.

### 3.2. Chemometric Analysis of the Relationship among Physicochemical Variables

The analysis of the correlation matrix revealed significant correlations among certain variables. Ash content showed a positive correlation with K, Na, Ca, pH, conductivity, resistivity, salinity, TDS, and total acidity (*p* < 0.05 for all; see Figure 1). Similarly, K exhibits a positive correlation with Na, Ca, conductivity, resistivity, salinity, TDS, and total acidity (*p* < 0.05 for all; see Figure 1). In contrast, Na displayed a positive correlation with salinity, conductivity, TDS, and K but a negative correlation with resistivity.

The linear relationship between the ash content and the electrical conductivity might be explained, since the higher the ash content, the higher the resulting conductivity recorded. This is the reason why the electrical conductivity is currently determined during routine honey control instead of the ash content [5]. Additionally, it is well known that the mineral content (such as Na, K, and Ca) and pH are closely related to the measured electrical conductivity [46].

Notably, resistivity and HMF are the only physicochemical parameters that showed a negative association with others (*p* < 0.05 for all; see Figure 1). On the other hand, the lactone acidity did not demonstrate any significant correlations with any other variables, and the moisture only showed a negative association with the °Brix values (*p* < 0.05; see Figure 1), which means that the °Brix values are inversely proportional to water contents. The relatively higher moisture content in the honey samples collected in April may have resulted from honey harvested under higher moisture conditions, or this can also be attributed to the botanical sources.

The FAMD analysis showed that components 1, 2, and 3, including all physicochemical parameters, explained 81.9% of the cumulative variance. Component 1 accounted for 55.1% of the total variance, and component 2 explained 17.1% of the variance, while component 3 explained 9.7% of the variance (Figure 2A). Salinity, conductivity, K, ash content, total acidity, pH, Ca, Na, HMF, free acidity, °Brix, TDS, and resistivity contributed significantly (*p* < 0.001 for all) to component 1, while lactone acidity, °Brix, Ca, HMF, and moisture had the strongest association (*p* < 0.001) with component 2. Lactone acidity, Brix, TDS, moisture and total acidity were strongly associated to component 3 (*p* < 0.001). The FAMD analysis revealed that the harvest period significantly contributed to components 1 and 2 (*p* < 0.01 for both), with the harvest period in April and February contributing to component 1, whereas the harvest period in March contributed to component 2. The HCPC analysis further supported the association between the physicochemical parameters and the harvest period during the dry season (Figure 2B), as indicated by a significant association between the harvest period and the clusters (*χ ^2^*; *p* < 8.54 × 10^−16^). Clusters 1, 2, and 3 were predominantly associated with the harvest periods of February, March, and April, respectively (Figure 2B).

Most of the physicochemical parameters exhibited significant variation across the harvest periods, with the exceptions of Na, pH, TDS, and lactone acidity (Appendix A). Notably, February and March showed significant differences across multiple physicochemical parameters (Table 3). This finding was consistent with the patterns observed in the HCPC analysis (Figure 2B) in which clusters 1 and 3 are distinctly separate (represented by circles and squares, respectively). These results highlight a distinct change in the physicochemical composition during this specific time frame.

The honey samples studied in this research project were multifloral. We found 92 pollen morphospecies from 30 plant families in the first harvest (February), 91 pollen morphospecies from 33 plant families in the second harvest (March), and 102 pollen morphospecies from 27 plant families in the third harvest (April). These findings reveal a higher pollen diversity than those reported by Roubik et al. from other regions of Panama [23]. The hierarchical cluster analysis revealed distinct patterns in the pollen species compositions among the honey samples harvested in February, March, and April (Figure 3 and Appendix A). Interestingly, the analysis demonstrated that the pollen species composition of the samples from March and April displayed greater similarity compared to the samples from February. This finding suggests a potential temporal shift in the dominant pollen sources or environmental factors influencing the floral composition during the harvest periods.

## 4. Conclusions

This study shows that the physicochemical characteristics and pollen composition of honey differ among different harvests of a single year at the same site, but we did not aim to assess the variation at the colony level. Some parameters may be of particular importance for local producers, enabling them to identify those times of the year at which to harvest the highest-quality honey. The results provide a clear example of the potential of scientific research to offer more and better tools to honey producers, allowing them to harvest higher quality honey with better nutritional value. The results also highlight one of the challenges faced by producers in some tropical areas, as honey harvested at the start of the rainy season had a higher water content. Honey harvested during this period may be more susceptible to fermentation and so may be of lower quality than that harvested through the previous months in the dry season (February and March).

## Figures and Tables

**Figure 1 foods-12-03656-f001:**
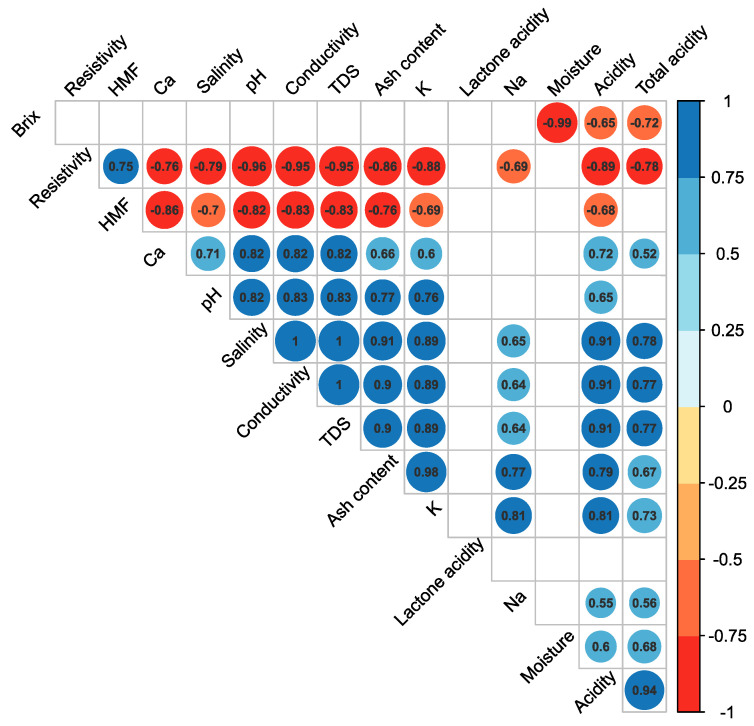
Honey bee physicochemical correlation matrix. Physicochemical data were analyzed by estimating the Spearman correlation coefficient for each pairwise comparison, which is displayed in each cell. Positive correlations (from 0 to 1) are displayed in blue, and negative correlations are in red (from 0 to −1). The color intensity (dark to light) and size of the circles in the cells (small to large) indicate the strength and direction of the relationships among the physicochemical parameters. The matrix only illustrates statistically significant values (*p* < 0.05), while nonsignificant correlations are left as blank cells.

**Figure 2 foods-12-03656-f002:**
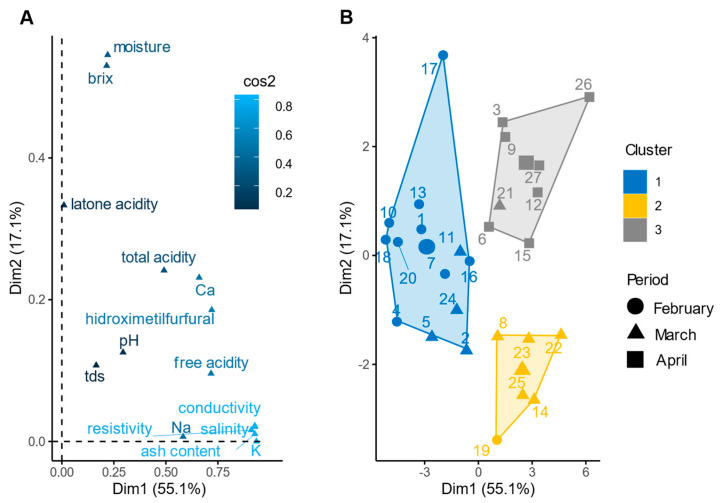
Relationship between the physicochemical parameters and harvest period of the honey samples: (**A**) factor map showing the physicochemical parameters represented by their quality in which a high cos2 score (lighter blue) indicates a good representation of the variable; (**B**) cluster analysis generated with HCPC showing the relationships among the physicochemical parameters during the dry season. The percentage next to the axis labels indicates the variance explained by each component, and the largest shape denotes the centroid for each group.

**Figure 3 foods-12-03656-f003:**
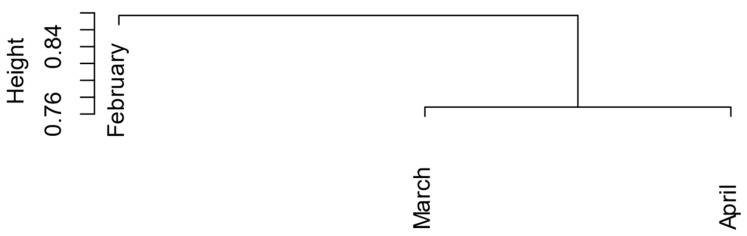
Pollen species diversity across the harvest periods of the honey samples. The clustering analysis, performed using the Ward method, reveals the grouping patterns and similarities among the pollen species. The cluster represents 217 morphospecies of pollen detected across honey samples.

**Table 1 foods-12-03656-t001:** Analytical methods and operating conditions used in the mineral determination.

Mineral	Method	Wavelength (nm)	Slit Width (nm)	Lamp Intensity	Calibration Curve linear Working Range (mg/L)	Quantification Limit ppm (µg/g) (Wet Mass)
K	Flame emission photometry	766.5	0.7	n/a	5–30	500
Na	Flame emission photometry	589	0.2	n/a	2–10	20
Ca	Atomic absorption spectroscopy (flame)	422.7	0.7	10	0.5–5	50
Pb	Atomic absorption spectroscopy (flame)	217	0.7	10	0.5–5.0	1.25
Cd	Atomic absorption spectroscopy (flame)	228.8	0.7	8	0.05–0.5	0.125

n/a = not applicable when flame emission photometry is performed.

**Table 2 foods-12-03656-t002:** Phytochemical parameters and minerals obtained from hives (mean ± SD).

Physicochemical Parameters	Period		
	February	March	April
Moisture (%)	17.91 ± 0.77	18.03 ± 1.87	20.93 ± 0.86
pH	4.29 ± 0.29	4.23 ± 0.13	4.33 ± 0.12
Free acidity (meq/kg)	33.02 ± 8.79	44.38 ± 8.23	51.66 ± 9.93
Lactone acidity (meq/kg)	23.59 ± 9.82	21.00 ± 2.30	23.15 ± 4.03
Total acidity (meq/kg)	56.66 ± 16.59	65.40 ± 9.18	74.82 ± 13.18
Ash content (%)	0.21 ± 0.05	0.32 ± 0.12	0.42 ± 0.13
Conductivity (mS/cm)	405.7 ± 166.87	574.85 ± 172.53	654.85 ± 127.42
°Brix	80.50 ± 0.97	80.17± 1.89	77.31 ± 0.79
TDS (mg/L)	202.87 ± 83.34	287.56 ± 86.28	327.38 ± 63.47
Resistivity (Ω⋅m)	5.08 ± 1.65	3.44 ± 0.98	2.87 ± 0.49
Salinity (psu)	0.20 ± 0.08	0.27 ± 0.08	0.31 ± 0.06
HMF (mg/kg)	49.49 ± 13.72	39.15 ± 9.37	38.73 ± 8.73
K (µg/g)	711.82 ± 307.41	1254.4 ± 417.48	1563.28 ± 471.42
Na (µg/g)	37.2 ± 17.47	46.96 ± 22.35	53.5 ± 17.24
Ca (µg/g)	71.84 ± 8.93	149.5 ± 61.24	135.56 ± 32.78
Pb (µg/g)	nd	nd	nd
Cd (µg/g)	nd	nd	nd

nd = Not detected.

**Table 3 foods-12-03656-t003:** Physicochemical parameters among different periods during the dry season. The statistical parameters according to the Kruskal–Wallis tests are shown in the last columns. Harvest periods that do not share a lowercase letter are statistically different, as determined using Dunn’s test for multiple comparisons with Bonferroni correction.

Physicochemical Parameter	Period			Kruskal–Wallis Test		
	February	March	April	*χ^2^ **	Df **	*p*-Value ***
Ash content	a	a,b	b	7.89	2	0.019
K	a	a,b	b	6.72	2	0.035
Na	na	na	na	2.66	2	na
Ca	a	b	b,c	9.38	2	0.009
pH	na	na	na	5.88	2	na
Conductivity	a	b	b,c	12.43	2	0.002
Resistivity	a	b	b,c	12.43	2	0.002
Salinity	a	b	b,c	12.35	2	0.002
TDS	na	na	na	1.34	2	na
Brix	a	a	b	13.66	2	0.001
Moisture	a	a	b	13.75	2	0.001
Free acidity	a	b	b,c	13.61	2	0.001
Lactone acidity	na	na	na	1.78	2	na
Total acidity	a	a,b	b,c	8.29	2	0.016
HMF	a	b	b,c	10.25	2	0.006

* Chi-square, ** degree of freedom, *** statistically significant (*p* ≤ 0.05); na indicates a statistically nonsignificant difference.

## Data Availability

The data are available from the corresponding author.

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
