# Peer review of "Assessment of the Quality, Chemometric and Pollen Diversity of Apis mellifera Honey from Different Seasonal Harvests"

_foods, 2023, doi:10.3390/foods12193656_

Round 1

Reviewer 1 Report (New Reviewer)

The article titled ‘Assessment of the quality, chemometric and pollen diversity analyses of Apis mellifera honey among different annual harvests’ provides detailed insights into the variation of physicochemical parameters of honey.

Plagiarism: 28 % reduce it up to a permissible limit

After a thorough reading, I have the following points of improvement for the following sections:

Abstract:

  • The abstract could be made more concise, focusing on the primary objectives, significant findings, and implications.
  • The abstract could emphasize the uniqueness of studying honey from the dry arc of Panama.
  • The concluding statement could specify how these results can guide honey production standards or practices in the region, making it relevant for stakeholders.

Introduction:

  • The structure of the introduction could follow a more logical progression. Start with the importance and uses of honey, its adulteration issues, and factors affecting honey quality, and then narrow down to the specific objective of the study.
  • Emphasize why understanding seasonal variation in Panama's dry arc is essential. This will highlight the significance of the study in the regional context.
  • Explicitly state the gaps in existing literature that this study seeks to address. This would help in justifying the need for the current research.

Methodology:

·        Clarify why only five beehives were chosen. Did these represent typical hives, or were they randomly selected?

·        Ensure that sample collection methods are consistent across all three harvest periods.

·        Elaborate on why the chosen 16 parameters are representative or critical for determining the quality of honey, especially in the given context.

·        For the uninitiated reader, briefly describe or provide references for methods like "Factor Analysis of Mixed Data (FAMD)" and "Hierarchical Clustering on Principal Components (HCPC)".

·        The procedure for handling missing data should be clearly described. Address why certain data might have been missing and ensure that imputation methods do not introduce bias.

·        Define acronyms before their usage, for instance, HCPC.

·        When choosing specific statistical tests (like Kruskal-Wallis), briefly justify their selection over others.

·        Emphasize the importance of pollen analysis. Why is understanding the floral composition crucial for the quality assessment of honey?

·        Ensure consistent sample sizes. If three out of five beehives were used for pollen analysis, explain the reason for this choice.

Results and Discussion:

·        The presentation of data in Table 2 is clear, but consider using visual aids like graphs or charts to help readers visualize the differences and trends in physicochemical parameters across different harvest months. Visuals can enhance understanding and engagement.

·        In the discussion of the results, provide more context and explanations for the observed variations. For example, why does the water content increase in April? Are there environmental factors that might explain these changes?

  • When discussing statistical results (e.g., Kruskal-Wallis tests), explain the significance of the findings in practical terms. What do these statistical differences mean for honey quality or production?
  • Provide more information about the implications of the mineral content findings. Discuss how the mineral content might affect the nutritional value or safety of honey, especially regarding heavy metals like Pb and Cd.
  • Explain in more detail the significance of HMF content as a freshness indicator. What are the potential health implications of higher HMF levels? Provide more context about the acceptable limits and regulations.

Conclusions:

  • In the conclusions section, summarize the main findings of the study. Highlight the key differences and trends observed in honey quality and pollen diversity across different harvest periods.
  • Discuss the practical implications of your findings for honey producers and consumers. For instance, how can honey producers optimize their harvest timing to ensure higher-quality honey? What can consumers learn from this study?
  • Suggest areas for future research. Are there unanswered questions or additional factors that could influence honey quality and diversity? Guiding potential follow-up studies can be valuable.
  • Offer practical recommendations based on your findings. These recommendations could be for beekeepers, regulators, or researchers in the field.
  • Discuss the broader significance of your study in the context of honey production, food safety, or beekeeping practices. Why is understanding the variations in honey quality important?

General:

  • Ensure consistency in the format. For instance, the spacing in the numbering (like 2.2.1. pH...), and the consistency in citing references ([24] and then [25]).
  • Use consistent terminology throughout the paper. For instance, if "honeybees" is used in one section, it should not appear as "honey bees" in another section unless denoting a different concept.
  • Ensure the language is formal and free from colloquialisms. Avoid using passive voice excessively.

Remember to ensure clarity and coherence in your writing, making it easy for readers to follow your article. Additionally, consider referencing relevant literature to support your interpretations and recommendations.

Minor editing of English language required

Author Response

Reviewer 2 Report (New Reviewer)

Abstract:

Page 1, L 27: Replace “in which” with “where”

Page 1, L 29: Replace “aimed” in “aims”

Introduction: 

Pages 1 and 2: LL 42-49. I suggest to replace this entire part with something more relevant. The reader knows what honey is. 

Page 2, L 50: Replace “Today” with “Currently” or a synonymous.

Page 2, L 51: Remove “In essence”. 

Page 2, LL 51-52: The sentence “Honey produced by honey bees is a mixture of sugars, especially fructose, and glucose” is not appropriate. Which sugars? The authors should report in the text the percentage of each component of honey. 

Page 2, LL52-52: “In addition, honey contains small amounts of phenolic acids, flavonoids, minerals, amino acids, proteins, vitamins, and essential oils”, same question of before. Percentage of each component should be indicated. 

Page 2, LL 63-65: “Honey adulterations can take place due to the substitution …”, I suppose the author would like to say “ by means of the substitution” or synonymous. As it has been written, it is not clear. 

Page 2, L 65: what other is for? Please, specify. 

Page 2, L 67: Replace “further” with “furthermore”

Page 2, LL 67-68: “honey can change properties due to heating or storage in unsatisfactory conditions”. Please, re-write. For example, “Honey’s properties could change/could be affected/… by several factors such as… “.

Page 2, L 72: “Higher” does not make sense. Higher than how much? 

Page 2, LL 79-80: Which factors? In this way, the reader cannot follow the story. Please, specify. 

Page 2, L 91: Some minerals? Which minerals the authors would like to refer to? 

Page 2-3, LL 94-96: This sentence is not appropriate here. Please, remove. 

Materials and Methods

Page 3, L 109: Please re-write. For example, Honey sampling, Sampling campaign of honey… 

Page 3, L 111: “We collected” is not appropriate. Re-write the sentence. 

Page 3, L 119: The sentence must be removed. This sentence is suitable for the introduction for example, but not for material and methods section. 

Page 3, L 120: “In order to know” should be replaced with something more formal. This sentence should be written as “ To assess/to determine the physiochemical characteristics.. “

Page 3, LL: 120-124: This sentence is really long and should be completely revised. It is not correct in English. 

Page 3, L 126: Accurately is not necessary. Please, remove it. 

Page 3, L 128: the solution without “resulting”. Homogenised for how many times? This section is titles “Materials and Methods”, therefore, all procedures followed should be described in detailed. 

Page 3, LL 128-132: the samples were measured is incorrect. What did the author measure? 

Page 3, L 134: “Were determined” sounds better. Please, correct it. 

Page 3, LL 138-144: Please, re-write.

Page 4, L 151: Please, be coherent throughout the text. “Five” should be written as a number (e.g. 5) 

Page 4, LL 150-158. The sentences are completely incorrect from a grammatical point of view. Please, check them and correct. 

Page 4, LL 186-200. Re-write. 

English need to be improved. The majority of the manuscript contains grammatical errors. 

Round 2

Reviewer 1 Report (New Reviewer)

Accept in present form

Author Response

We would like to thank the reviewer for the careful and thorough reading of this manuscript and for his/her comments and constructive suggestions, which helped to improve the quality of this manuscript. 

Reviewer 2 Report (New Reviewer)

Revision

Line 58: Define “High quality”.

Line 61-63: Please, for each method, add a reference.

Line 77: Replace “objective” with “aim” or “scope” or “purpose”.

Titles of paragraph should be write in italics.

Author Response

Point 1: Line 58: Define “High quality”.

  • Response: High quality is defined as follows: "Honey intended for human consumption should be of high quality, that is safe and free of contamination and defects". 

Point 2: Line 61-63: Please, for each method, add a reference.

  • Response: Each method has been referenced in the text.

Point 3: Line 77: Replace “objective” with “aim” or “scope” or “purpose”.

  • Response: Objectives is replaced with aim, as suggested by the reviewer.

Point 4: Titles of paragraph should be write in italics.

  • Response: Titles of paragraphs have been written  in italics as suggested by the reviewer.

This manuscript is a resubmission of an earlier submission. The following is a list of the peer review reports and author responses from that submission.

Round 1

Reviewer 1 Report

Dear authors,

Modern research focuses on assessing the quality of natural bee honeys, as well as on the search for markers and distinguishing factors.

The subject matter is interesting, but in my opinion, research should be carried out on a larger scale, honey should come from many different hives.

Chemometric analyzes are used, among others, to finding dependencies, rules and reducing the amount of data.

Author Response

Point 1: Modern research focuses on assessing the quality of natural bee honeys, as well as on the search for markers and distinguishing factors.

Response 1: We share the reviewer's idea about chemometric studies and the other general comment about “modern research.”

Point 2: The subject matter is interesting, but in my opinion, research should be carried out on a larger scale, honey should come from many different hives.

Chemometric analyzes are used, among others, to finding dependencies, rules and reducing the amount of data.

Response 2:  In the previous review we explained that the objective of the manuscript was to identify if there was a difference between the different harvests of the same year, and we explained that our study began with 22 hives, but only 5 hives produced the minimum amount of honey from ours (> 2.5 gallons to have a representative sample of the hive per harvest). To our knowledge, this is the first publication of its kind for the tropics and it is important for quality strategies in honey production.

We hope that you agree with your editorial colleague that our manuscript is ready to be published in Foods.

With best Regards

Reviewer 2 Report

As a reviewer of the previous version of the manuscript, as well, I could see a definite improvement in the quality of the current manuscript. The authors have addressed all the issues raised by me in my previous comments.

The discussion has been extended, providing explanations for what might be in the background of the differences detected between honey samples / harvest periods.

As suggested previously, the authors' own findings have been contrasted with those of other research groups and international regulations regarding honey quality were taken into consideration.

I think that it is an interesting study, with practical / economic relevance, which can be now accepted for publication.

Author Response

On behalf of my co-authors, I am pleased to thank reviewer 1 for the careful and thorough reading of this manuscript and for their comments and constructive suggestions, which helped to improve the quality of this manuscript.

We agree with reviewer 1, so we have followed and made each of the recommendations and suggestions that he indicated to us.

We are grateful for your consideration.

Reviewer 3 Report

Accepted

Author Response

On behalf of my co-authors, I am pleased to thank reviewer 2 for the careful and thorough reading of this manuscript and for their comments and constructive suggestions, which helped to improve the quality of this manuscript.

We agree with reviewer 2, so we have followed and made each of the recommendations and suggestions that he indicated to us.

We are grateful for your consideration.

Reviewer 4 Report

The manuscript concerns the evaluation of honey quality harvested in different periods in the region of Panama with low precipitation. Results and discussion section is lack of references comparing the results with other studies. Different references should be added if possible. Other comments are listed below:

L32: ‘increases during the harvest period from February to April’ – increases how many %

L33: indicate the main results of PCA

L52: add also that honey contains royal jelly aliphatic acids. Some indicated references in this sentence are quite old. Replace them with newer e.g. https://doi.org/10.2478/jas-2018-0012, https://doi.org/10.3390/ijerph20032458, https://doi.org/10.1016/j.jfca.2021.103837

L107: what were the types of honey in terms of botanical origin?

L121: Brix – add a full name

L185: indicate in statistical analysis that spearman correlations for correlation matrix were performed

L241: add the examples of these honeys

L268: it is 27.2 meq/kg in Table 2

L284: the lowest value is 236 mS/cm in Table 2

L290: add unit for values

L340-345: add some values of correlation

L372-373: component 3 is not shown in Fig. 2

L390: Fig. 2B instead Fig. 4B

Author Response

  • The manuscript concerns the evaluation of honey quality harvested in different periods in the region of Panama with low precipitation. Results and discussion section is lack of references comparing the results with other studies. Different references should be added if possible.

Reply: We believe that the approach of this research article is to help as a first step in characterizing the quality of honey produced in Panama. Our research group is currently working with honey collected from different areas in the country so we can make a more appropriate comparison with the findings obtained from other tropical regions in the near future. So, we do think that, at this research point, is more appropriate to compare our results referring them to the Codex Alimentarios, EU Directive and Panamanian legislation.

Other comments are listed below:

  • L32: ‘increases during the harvest period from February to April’ – increases how many %

Reply: The text has been modified as follows: …. its concentration increased by 120% during the harvest period from February to April (L34).

  • L33: indicate the main results of PCA

Reply: We used a Factor Analysis of Mixed Data (FAMD), which incorporates quantitative (PCA) and qualitative (MCA; multiple correspondence analysis) variables (L34). Quantitative and qualitative variables are normalized to balance the influence of each set of variables. The analysis is described in lines 196 to 198 and the results are described in lines 371 to 385.

  • L52 (now L55): add also that honey contains royal jelly aliphatic acids. Some indicated references in this sentence are quite old. Replace them with newer e.g. https://doi.org/10.2478/jas-2018-0012, https://doi.org/10.3390/ijerph20032458, https://doi.org/10.1016/j.jfca.2021.103837

Reply: We have added newer references as recommended by the reviewer (see L56 and references 3-8.

  • L107: what were the types of honey in terms of botanical origin?

Reply: It was not determined but is very important to conduct future scientific studies regarding this topic.

  • L121: Brix – add a full name

Reply: The full name has been added: total soluble solids (°Brix) (L123).

  • L185: indicate in statistical analysis that spearman correlations for correlation matrix were performed

Reply: The result of the statistical analysis is summarized in Figure 1, and the interpretation of the results is in lines 341 to 361.

  • L241: add the examples of these honeys

Reply: We added examples of honey as follows: the pH values of different types of blossom honey (linden, buckwheat, rapeseed, and acacia) usually range from 3.5 to 5.5 [32](L242-L243).

  • L268: it is 27.2 meq/kg in Table 2

Reply: The highest value has been corrected (L270).

  • L284: the lowest value is 236 mS/cm in Table 2

Reply: The lowest value has been corrected (L286).

  • L290: add unit for values

Reply: We have added the unit values as follows: 76.5 to 82.0 % (L292).

  • L340-345: add some values of correlation

Reply: The summary of the Spearman correlation coefficient for each pairwise comparison is annotated in Figure 1. “Color intensity (dark to light) and size of the wells (small to big) indicate the strength and direction of the relationship between the physicochemical parameters.”

  • L372-373: component 3 is not shown in Fig. 2

Reply: Figure 2A represents a factor map that shows the representation of each physicochemical parameter. Components 1 and 2 are used in the figure because they account for most of the variation. 

  • L390: Fig. 2B instead Fig. 4B

Reply: The name of the figure has been corrected (L392).

Round 2

Reviewer 4 Report

The authors have corrected the paper according to most of the comments. I have no more questions.

Author Response

On behalf of my co-authors, I am pleased to thank you for your comments and the opportunity to revise our manuscript. Your constructive suggestions led us to improve our research article.